# High-resolution spatiotemporal measurement of air and environmental noise pollution in Sub-Saharan African cities: Pathways to Equitable Health Cities Study protocol for Accra, Ghana

Sierra N Clark [ID],[1,2] Abosede S Alli,[3] Michael Brauer,[4] Majid Ezzati,[1,2,5,6] Jill Baumgartner,[7,8] Mireille B Toledano,[1,2] Allison F Hughes,[9] James Nimo,[9] Josephine Bedford Moses,[9] Solomon Terkpertey,[9] Jose Vallarino,[10] Samuel Agyei-Mensah,[11] Ernest Agyemang,[11] Ricky Nathvani,[1,2] Emily Muller,[1,2] James Bennett,[1,2] Jiayuan Wang,[3] Andrew Beddows,[2] Frank Kelly,[2,12] Benjamin Barratt,[2,12] Sean Beevers,[2] Raphael E Arku[3]

For numbered affiliations see end of article.

**Correspondence to**
Dr Raphael E Arku;
rarku@umass.edu

## ABSTRACT

**Introduction** Air and noise pollution are emerging environmental health hazards in African cities, with potentially complex spatial and temporal patterns. Limited local data are a barrier to the formulation and evaluation of policies to reduce air and noise pollution.

**Methods and analysis** We designed a year-long measurement campaign to characterise air and noise pollution and their sources at high-resolution within the Greater Accra Metropolitan Area (GAMA), Ghana. Our design uses a combination of fixed (year-long, n=10) and rotating (week-long, n =~130) sites, selected to represent a range of land uses and source influences (eg, background, road traffic, commercial, industrial and residential areas, and various neighbourhood socioeconomic classes). We will collect data on fine particulate matter ($PM_{2.5}$), nitrogen oxides ($NO_x$), weather variables, sound (noise level and audio) along with street-level time-lapse images. We deploy low-cost, low-power, lightweight monitoring devices that are robust, socially unobtrusive, and able to function in Sub-Saharan African (SSA) climate. We will use state-of-the-art methods, including spatial statistics, deep/machine learning, and processed-based emissions modelling, to capture highly resolved temporal and spatial variations in pollution levels across the GAMA and to identify their potential sources. This protocol can serve as a prototype for other SSA cities.

**Ethics and dissemination** This environmental study was deemed exempt from full ethics review at Imperial College London and the University of Massachusetts Amherst; it was approved by the University of Ghana Ethics Committee (ECH 149/18-19). This protocol is designed to be implementable in SSA cities to map environmental pollution to inform urban planning decisions to reduce health harming exposures to air and noise pollution. It will be disseminated through local stakeholder engagement (public and private sectors), peer-reviewed publications, contribution to policy documents, media, and conference presentations.

## Strengths and limitations of this study

► Our study is the largest air and noise pollution measurement campaign conducted in a Sub-Saharan African city and serves as a prototype for other cities in SSA.

► The study relies on new sensor technologies to generate rich datasets on air and noise pollution along with imagery and audio recordings that help identify sources across ~140 locations.

► Data from a combination of fixed (1 year, n=10) and rotating (7 days, n=~130) monitoring sites representing a diversity of areas will allow for an assessment of both the temporal and spatial variability of pollution.

► While our study makes use of next-generation low-cost technologies, significant need for human resources is required for site identification and preparation, equipment deployment and maintenance, and data download and management.

## INTRODUCTION

Sub-Saharan Africa (SSA) is the world's fastest urbanising region, with the number of urban dwellers having increased by over 400% from 84 million in 1980 to an estimated ~450 million people in 2020.[1] Urban growth in SSA has been largely unplanned especially in relation to housing, transport and energy. As a result, air and noise pollution are increasingly a public health concern for SSA urban residents.[2–4] For example, estimates from global models suggest that ambient fine particulate matter (particulate matter with diameter <2.5 micrometres, $PM_{2.5}$) in SSA is well above levels in high-income North

America and Western Europe.[3 5] The data from the few available measurement studies show that only about 10% of cities in SSA are meeting the WHO annual average Air Quality Guideline of 10 µg/m[3].[5 6] While such global estimates and the limited measurement data provide a broad view of air pollution, they do not capture the spatial variability and within-city disparities, nor do they provide information on sources.[7–9] Those within-city differences are important determinants of pollution-related health inequalities. There are even less data on noise pollution, and none on its health burden. The limited noise data show much higher levels compared with cities in high-income countries,[10–15] which may be associated with hearing loss, sleep disturbance, impaired cognitive function, and cardiovascular disease.[16–18]

Air and noise pollution in SSA have a complex mix of local and regional sources: these include informal industries; transportation predominantly from old imported vehicles for commercial and private use; biomass use for household and commercial activities; household trash burning; resuspended dust from unpaved roads; dust from regional dust storms; and noise from road traffic, small road-side businesses, and religious practices, to name a few.[4 7 9 19 20] These sources influence the pollutant mixture and the type of urban sounds, resulting in variation in spatial patterns and potentially differential impacts on health. Carefully designed measurements using low-cost robust sensors present an opportunity to provide data on the levels, variations, and sources to inform and evaluate the effectiveness of policies in SSA.

Motivated in part by earlier air pollution data from four neighbourhoods in the city core, Accra, Ghana's largest city, in 2018 announced initiatives to reduce air pollution,[21] whereas noise is currently making headlines in both local and international media.[22–24] Our goal is to leverage advancement in sensor technology, modelling and image processing to design a measurement campaign combined with machine learning, statistical and process-based modelling to characterise highly resolved space-time variability of air and noise pollution, and their sources in the Greater Accra Metropolitan Area (GAMA). This work is nested within the larger multicountry and multicity 'Pathways to Equitable Healthy Cities' study (http://equitablehealthycities.org/), which aims to identify and inform equitable and healthy urban development and revitalisation pathways in six cities on four continents.

This paper details the protocol being used to collect and analyse large-scale pollution data in high resolution and provides practical guideline in a rapidly growing SSA metropolitan area. As one of the few studies of air and noise pollution at fine spatial resolution in a SSA city, this paper and the data to be generated make three main contributions. First, to develop and implement a data-rich measurement campaign on air and noise pollution in the GAMA that can provide spatially and temporally graded data. Second, to present a measurement protocol that can be readily adapted to other SSA cities. Third, to describe how the data will be used to fit and/or validate

geostatistical, machine learning and physical dispersion models that can predict pollution levels at high spatial and temporal resolution and simulate and evaluate different policy scenarios on air quality in Accra.

## METHODS AND ANALYSIS
### Study location and timeline
Our measurement campaign is focused on the GAMA, which covers about 1500 km[2], and consists of multiple metropolis and municipalities, with Accra Metropolitan Area (AMA) at its core (figure 1). Accra lies in the dry equatorial climate zone with rainy (May–September) and dry Harmattan seasons characterised by dusty north-easterly trade winds from the Sahara Desert. The elevation of the GAMA is near sea level. Monthly average temperatures range from 27°C to 32°C with average daily humidity of 79%.[25] As Ghana's capital and largest city, Accra has become one of SSA's hubs for business, technology, communications and education. However, there remain large inequalities in housing and possibly exposure to environmental health risks.[8 26–28]

We scheduled a 1-year field measurement campaign to cover the rainy and Harmattan periods. Measurements began with a 3-week long pilot campaign in April 2019 and will continue until May 2020 (figure 2).

### Measurement campaign design
To capture the temporal (daily, weekly, seasonal) and spatial variations in both pollution and its sources across the entire study area, we are using a combination of 'fixed' and 'rotating' monitoring sites. The sites represent a blend of features such as background areas (eg, low traffic and high green space), low versus high road traffic, sparsely versus densely built-up areas, poor versus affluent and established versus emerging neighbourhoods.

Ten fixed sites have been installed and will operate continuously all year long; the sites were purposefully selected based on the above criteria related to population density, road traffic and road networks, and on neighbourhood socioeconomic status and biomass fuel use based on national census data.[29] The sites included four locations used in an earlier air pollution study[7 26] in the AMA and additional provisions have been made to colocate with World Bank sponsored regulatory monitoring sites and one at the US Embassy.

To capture spatial patterns of pollution while maximising a finite number of sensor packages, we also operate sites that rotate weekly in order to capture the spatial variation in pollution levels and sources as well as the temporal variation within and between days. In each measurement week, measurements are collected at four to five new locations that continuously monitor for 7 days. By the end of the study, ~130 unique locations will have been monitored for 1 week across the GAMA.

In selecting the rotating site locations, we used a stratified random sampling approach as follows:

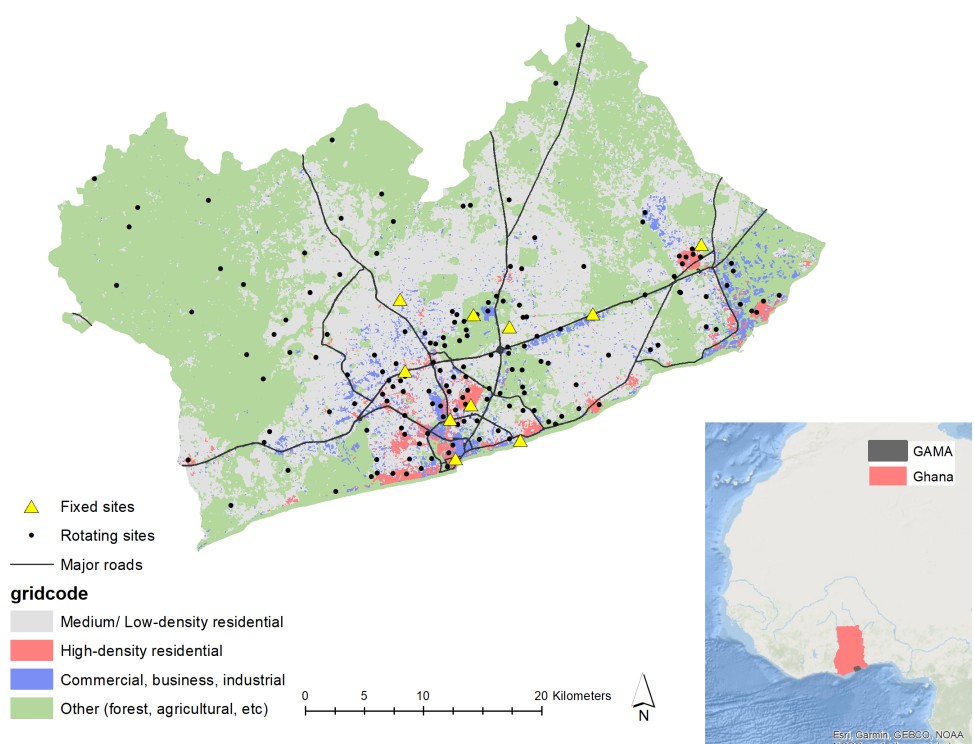

**Figure 1** The Greater Accra Metropolitan Area (GAMA) and locations of the fixed and computer-generated (sampled) rotating sites. The road network data are from OpenStreetMap and the background land cover shapefile is from the World Bank (2014). The inset shows background maps of Africa and Ghana (ESRI (Environmental Systems Research Institute)), along with the GAMA boundary from Ghana Statistical Service. High-density residential indicates neighbourhoods with small, crowded, irregular buildings and narrow unidentifiable unpaved roads such as in shanty towns and slums. Medium/low-density residential indicates neighbourhoods with small regular planned buildings and indicate formal residential areas. Commercial/ business/ industrial indicates neighbourhoods with large buildings that can be used for commercial, industrial, office or warehouse purposes. Other indicates areas with large spaces of vegetation (eg, dense forest), barren land (eg, sand, soil) or water bodies.

1. The study area (GAMA) was stratified by a land use grid (20 m × 20 m raster converted into a polygon shapefile) with four classes (medium/low-density residential; high-density residential; commercial, business and industrial areas; and 'other' areas (eg, parks, forest, agricultural areas))[30] and inside or outside AMA.
2. The computer then generated and returned the latitude/longitude coordinates of a random sample of target measurement locations within strata.
3. Target measurement locations were first examined by overlaying point locations onto Google Maps and Google Earth to identify sites that were in restricted areas (eg, military barracks). Sites in restricted areas were re-sampled to a nearest suitable spot that also fell within the same type of land use strata (n=~5 sites).
4. Using the coordinates of the target sampling locations, the field team then visit individual sites throughout the campaign to find measurement sites at or as close as possible to the target locations and also with the same land use characteristics.
5. When a site is deemed structurally sound by the field team (eg, staircase to the roof) and can allow for the equipment to be installed at a target height, permission is requested from the site owner/manager (more details on the logistics of field work are in the next sections).

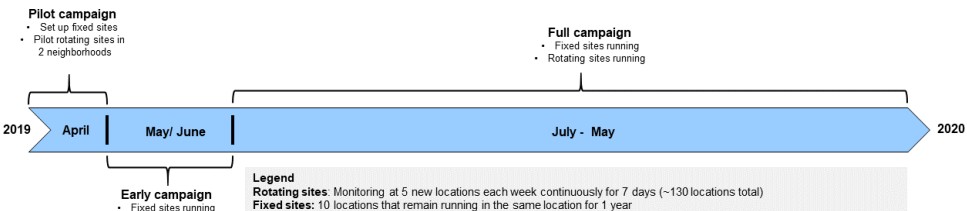

**Figure 2** Timeline of measurement campaign. Weekly measurements consist of continuous ($PM_{2.5}$ air concentration, noise levels, meteorological conditions, audio, and imagery) and integrated ($PM_{2.5}$ and $NO_x$ concentration) samples. We chose weekly integrated samples for $PM_{2.5}$ and $NO_x$ for logistical reasons (cost and time) as well as lessons from a previous study that showed relatively high temporal correlation between daily measurements.[8]

6. During the measurement campaign, we will actively review the balance between the number of actual measurement sites by land use strata as originally designed, and potentially sample additional sites to make up for unrepresentative site types.

## Measurement methods and equipment

We systematically selected and are employing low-cost, low-power and lightweight monitors that are robust and able to function in an environment characterised by high temperatures and humidity, rain and dust storms, and with limited and intermittent electricity supply from the grid, and at the same time are socially unobtrusive (table 1, figure 3).

### Air pollution monitors

*Integrated PM$_{2.5}$*: the Ultrasonic Personal Aerosol Sampler (UPAS)[31] from Access Sensor Technologies (Fort Collins, USA; UPAS) is a time-integrated PM$_{2.5}$ monitor and has a quiet solid-state miniature piezoelectric pump for drawing air through a customised cyclone onto a 37 mm diameter filter media contained in barcoded cartridges within the device. With a mass flow sensor and controller, UPAS provides a steady flow rate over time. A mobile app makes UPAS easily programmable to collect samples at varying duty cycles. The UPAS devices are being operated at 1 L/min at 50% duty cycle to avoid overloading the weekly integrated filters. The UPAS has been evaluated in laboratory and field settings against a federal reference monitor (URG-2000-30EGN-A; URG Corp., USA), personal environmental monitor (PEM 761-203; SKC, USA) and Harvard Impactors, respectively, and has proven valid for ambient, household and personal monitoring in a typical tropical climate as our study.[31–33]

*Continuous PM$_{2.5}$*: the ZeFan continuous monitor (http://www.zfznkj.com/) is a portable direct-reading PM$_{2.5}$ monitor that is based on light scattering technique.[34] ZeFan uses the Plantower sensor (model PMS7003) which has been validated against a TEOM 1400a analyser and tested for durations ranging from 6 months to a year in various environmental conditions.[34 35] Prior to field deployment, we tested minute-by-minute monitor-to-monitor precision by running 15 monitors alongside each other over a 24-hour period at the University of Ghana, Legon campus with average relative humidity (RH) (~78%) and temperature (29°C) representative of the city, and the measurements had good agreement (figure 4). Since light-scattering techniques only infer PM mass from detecting particle number concentrations and are impacted by weather conditions (ie, RH and temperature), their estimates of mass concentration are inexact. Thus, we will colocate the ZeFan with a US federal equivalent continuous monitor Met One BAM 1020 at three sites, each with unique source influence in Accra for a week at the end of the campaign and adjust the minute-by-minute PM records for impact of RH and then their average against the colocated integrated PM$_{2.5}$ concentrations from the UPAS.

**Table 1** Features, dimensions and prices of the monitors/sensors

| Monitor | Cost per unit (US$) | Weight (g) | Dimensions (cm) | Battery/power requirements | Memory requirements | Recording/ measurement interval | Measured parameters |
|---|---|---|---|---|---|---|---|
| Ultrasonic Personal Aerosol Sampler (UPAS)* | 1200 | 230 | 12.8×7.0×3.3 | Internal chargeable battery* | Micro SD | 7 days | PM$_{2.5}$ integrated (µg/m³) |
| ZeFan continuous PM$_{2.5}$ monitor* | 70 | 150 | 10.6×6.3×2.6 | Internal chargeable battery* | Internal memory (USB connection) | 1 min | PM$_{2.5}$ continuous (µg/m³) |
| Ogawa nitrogen dioxide (NO$_2$/ NO$_x$) sampler† | 85 | 60 | 8.0×4.0×3.0 | NA | NA | 7 days | NO$_2$ (ppb) integrated; NO$_x$ (ppb) integrated |
| Noise Sentry sound level meter | 306 | 100 | 7.6×3.9×5.9 | Internal chargeable battery | Internal memory (USB connection) | 1 min | Sound levels (dBA) |
| AudioMoth audio recorder | 70 | 95 | 6.2×5.0×2.2 | AA batteries | Micro SD | 10 s every 10 min | Audio (.WAV file) |
| Kestrel weather meter | 310 | 120 | 12.7×4.5×2.8 | AA batteries | Internal memory (USB connection) | 1 min | Temperature; relative humidity; wind speed; wind direction |
| Moultrie camera trap | 150 | 500 | 13.1×8.1×6.6 | AA batteries | SD | 5 min | Time-lapse imagery (.jpeg file) |

*UPAS and Zefan battery life can be extended using an external power bank. We used the always-on battery pack from Voltaic Systems (www.voltaicsystems.com).
†NO$_2$/NO$_x$: nitrogen dioxide/oxides (price includes clip, screens, plastic resealable pouch and reusable airtight storage and shipping vial).
dBA, A-weighted decibels; PM$_{2.5}$, particulate matter with aerodynamic diameter less than 2.5 micrometres; ppb, parts per billion.

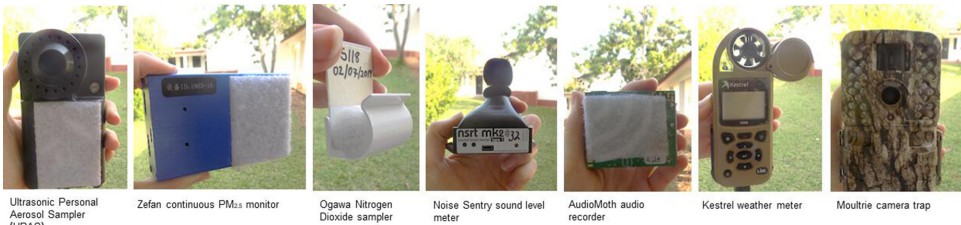

**Figure 3** Images of environmental monitoring equipment.

*Nitrogen oxides ($NO_x/NO_2$):* the Ogawa Passive Sampler (https://ogawausa.com) is being used to measure $NO_x$ and $NO_2$, which are inorganic gaseous indicators of traffic related air pollution.[36] The sampler is easy to deploy, reusable and does not require electricity, thus making it a cost-effective option in SSA settings. The sampler consists of two chambers with double-sided diffusion that can concurrently capture $NO_x$ and $NO_2$ concentrations on collection pads precoated with 2-phenyl-4,4,5,5-tetramethylimidazoline-1-oxyl-3-oxide and triethanolamine, respectively. The samplers are covered by an opaque plastic container which serves as weather shield.

### Sound monitors
*Sound levels:* the Noise Sentry Sound Level Meter (SLM) datalogger (NSRT_mk3) from Convergence Instruments, Canada, is being used to measure sound levels at 1 min integrating and logging intervals. The Noise Sentry is a relatively low-cost SLM with Type I precision for capturing and constructing common metrics of environmental noise pollution with multiple weighting curves. It is small and rugged, built to withstand temperatures in the range of −20°C to 60°C, and protected against water and dust. Previous studies have used the Noise Sentry SLM in diverse settings.[12 14] Our prepilot tests of monitor–to–monitor precision showed good agreement (more details in online supplementary information 1 (SI 1)). Our Noise Sentry SLMs were also validated in a separate aircraft noise study conducted in San Francisco against a Type I industry standard instrument (DUO 01 dB),[37] and the agreement was high (mean and median second by second difference between the instruments was −0.42 and −0.38 dBA, respectively).

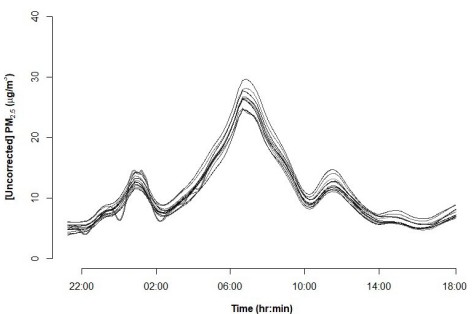

**Figure 4** Smoothed time series of minute-by-minute $PM_{2.5}$ from 15 colocated real-time Zefan monitors in Accra. The levels were neither corrected for relative humidity nor against integrated filter-based data.

*Audio:* the AudioMoth audio recorder is a low-cost, full spectrum, acoustic logger developed by Open Acoustic Devices (Oxford, UK).[38] The AudioMoth will complement the SLM by recording audio which will be used to classify different types of sounds in an urban environment (eg, animals, vehicle sounds). The AudioMoths are set to a sampling rate of 32 kHz in our study to capture the majority of sound in the audible range.[39]

### Weather monitors
The Kestrel 5500 weather metre (Nielsen-Kellerman, Boothwyn, Pennsylvania, USA) is being used to record weather variables every minute. The Kestrel is a handheld environmental metre and considered tough and immune to the elements. It tracks several weather parameters, including temperature, RH and heat index. It was selected for its low power consumption, large memory capacity (>10 000 data points), and dust-proof and waterproof properties. Kestrel 5500 has been used in several studies in diverse settings.[40 41] According to factory specifications, the accuracy of the instrument is 0.5°C for temperature and 2% for RH.

### Time-lapse cameras
To characterise sources of pollution in space and time, we use weatherproof and rugged time-lapse cameras (Moultrie-50 camera trap, PERDIX wildlife, UK). The cameras are programmed to capture images at 5 min intervals throughout the sampling period, including at night using infrared technology. Depending on the location, one or two cameras are mounted to capture multiple frames of view of potential pollution sources in the street such as cars and community cookstoves.

### Integrated equipment monitoring box
To house the equipment, we built integrated field measurement boxes using weather protective Seahorse (SE-300) cases. The cases were designed and weather tested to securely house each piece of equipment along with battery packs inside a single compartment, and could be mounted on poles of different sizes using ratchet straps. The cameras are mounted on the outside of the box with rotational multiaccess brackets for ease of orientation. Additionally, soundproof foam was placed in-between the air monitors and the SLM to mitigate internal sound that might be generated from the quiet air pollution monitor pumps. $NO_x/NO_2$ passive samplers and the audio recorders are placed outside of the measurement

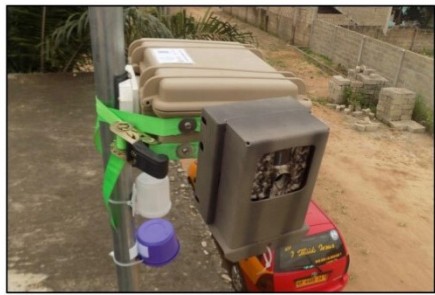
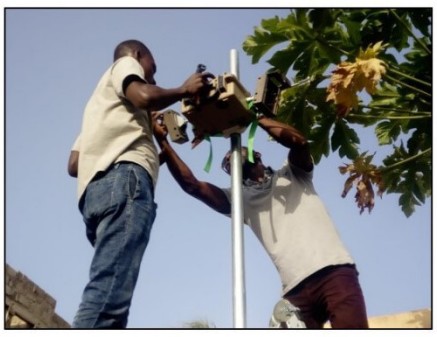
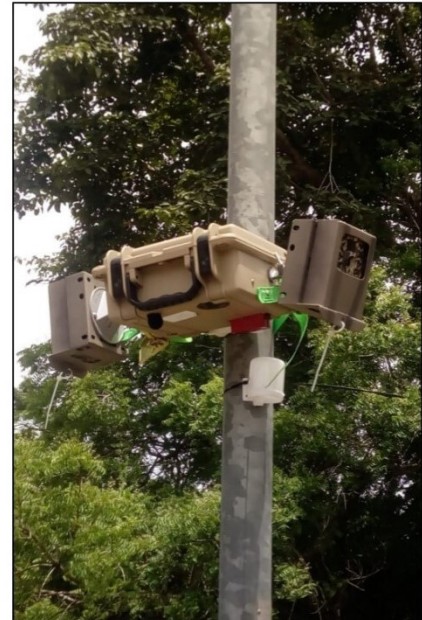

**Figure 5** Deployment of the pollution measurement equipment.

boxes in their own smaller weather protective plastic cases.

### Equipment deployment and data capture

The field team identifies potential sites at or as near as possible to the computer-generated locations using direction from the saved locations on Google Maps. The team then approaches residents, owners or managers, and explains the study rational and seeks approval to instal equipment for a 1-year (fixed sites) or 7-day (rotating sites) period. The team carry signed letters containing description of the research and contact information of project investigators at the University of Ghana. The site is then prepared, and the equipment box is mounted on metal poles, in care of an established contact person, and out of direct reach of passers by. Depending on the specific site, the poles are secured on flat rooftops/balconies of one-story buildings or directly in the ground (figure 5) about 4 m (±1 m) high, as is a common practice in ambient air pollution and noise measurement,[42] and also has no obstruction between the monitors and the sources of air and noise pollution. The cameras are mounted on the outside of the box and secured in metal cases.

After deployment, the field team completes a short form, documenting information about the site, including the presence or absence of visible pollution sources (eg, road-side cooking), mitigation factors (eg, trees) or other locations/features of interest, such as road-side food sales, shopping centres, schools, or hospitals. For the rotating sites, four to five locations are monitored each week. Because of logistical and time constraints related to setting up each site, the team chooses sites that are within the same part of the city, but may have varying land use characteristics (eg, mix of low and high-density

residential locations). Monitors are retrieved 7 days after initiating the measurements for data download and equipment cleaning in the field laboratory. The monitors are then redeployed 48 hours later at a new set of locations in a different geographic area, with the aim of capturing potential microclimate and source-related differences between areas which likely impact pollution. For the fixed sites, replacement monitors, replacement batteries and memory cards are swapped on site so as not to have a disruption in monitoring.

### Logistics and training

Our local field team comprises three recent graduates from the University of Ghana and a taxicab driver, all with technical training needed to manage the field operation. The team is given project specific training to understand the site selection criteria and to collect high-quality data. Additionally, periodic field visits and regular phone calls by researchers are made to maintain high-quality data. In each neighbourhood or community, the team identifies and works with a community member to establish trust and facilitate entry into that community.

### Data handling

Weekly data are downloaded, saved in triplicate onto two external hard drives, and a third copy uploaded to a sever at Imperial College via an encrypted laptop. For the integrated $PM_{2.5}$ filter samples, prelabelled 0.2 µm pore size 37 mm barcoded Teflon membrane filters (https://mtlcorp.com/filters/) are used and weighed pre- and post-sampling using an MTL AH500 automated robotic scale (http://www.mtlcorp.com/#/filter-weighing/) maintained in a temperature and RH-controlled laboratory (23°C±2°C, 35±2% RH) at The University of British Columbia. The filter labels are scanned, time stamped,

placed in individual carriers and loaded into the input silos for 48 hours to equilibrate to laboratory conditions before weighing. System-generated weighing reports (eg, balance stability) for each filter are issued for quality control purposes. Samples are weighed thrice in both pre- and post-weighing and the average of the three measured masses is used for calculating concentrations. The preweighed filters are scanned and paired to and placed in labelled petri dishes which are then sealed into individual packages. Each petri dish has four labels used to match the filter to the cartridge, UPAS monitor, and field log sheet during field work. After sampling, the filters are matched to, and placed back in, their corresponding petri dishes and shipped to the laboratory for postweights. Detailed information on the filter handling process can be found elsewhere.[32] An emerging low-cost image-based approach will be applied to the postweighed filters to estimate optical reflectance as a measure of black carbon (BC) concentration,[43] the mass related to light absorption due to the presence of carbonaceous species.

$NO_x/NO_2$ samples are handled according to the protocol from Ogawa.[44] After assembly in the laboratory, the loaded samplers contained in an airtight container are exposed only on site for the entire sampling week. After sampling, the above procedure is again followed, and samples refrigerated until the exposed pads are shipped in airtight shipping vials for laboratory analysis. The final sample concentrations are determined based on the ratio of the sample absorbance (measured by spectrometer) to the slope of a prepared standard curve. The full analytical method is publicly available.[44]

## Quality assurance and control
Throughout the campaign, we will follow a set of procedures and protocols to uphold and assess the quality of the data being generated. We follow the principles that all procedures should be carefully planned, tested and performed, the origin and life course of all data must be traceable, and any deviations or irregularities must be recorded. Throughout all data collection, documentation of sampling and conditions will be maintained in field notebooks. Furthermore, data collection logs will be digitised and backed up electronically on hard-drives and an online server, which will be checked on a daily basis for accuracy. The field team were given multiple weeks of project specific training prior to the pilot measurements. The team were taught specific protocols for equipment handling and cleaning, data inspection and cleaning, and equipment installation at measurement sites. The team were also given hardcopies of the protocols and, in addition to field visits by researchers, had constant remote access via phone/web to project researchers throughout the campaign. In the online supplementary information, we have included further information on our precision and accuracy testing, protocol for blank and duplicate collection, and data cleaning and inspecting procedures (SI 1).

## Modelling and analysis
The data from this measurement campaign will be used to characterise the spatial and temporal patterns of air and noise pollution and serve as inputs into a diverse suite of state-of-the-art statistical, physical and machine learning models to (1) predict pollution levels in high spatial and temporal resolution across the GAMA, (2) identify sources of pollution, and (3) simulate the impacts of policy scenarios on air pollution levels. Below are brief descriptions of some of our planned analysis and modelling activities following data collection. Future results-based papers will describe the modelling approaches in greater detail.

### Descriptive summaries of the spatial and temporal variations in air and noise pollution
We will provide summary statistics and visuals of the spatial and temporal patterns (within and between day, and seasonal) of air pollution ($PM_{2.5}$, $NO_2$) concentrations and average-based metrics of noise pollution such as $LAeq_{24hr}$, daytime ($L_{day}$), nighttime ($L_{night}$), and day–evening–night weighted $L_{den}$. Additionally, we will include metrics which capture short term and episodic sound events such as the maximum sound level and a novel metric that captures the percentage of event-based sound (the intermittency ratio ($IR_{24hr}$, $IR_{day}$, $IR_{night}$)).[45]

### High-resolution modelling of air and noise pollution in the GAMA
The increasing availability of geospatial datasets with land use characteristics,[46] road network information[47] and locations of interest (eg, locations of schools and hospitals) supports the development of land use regression (LUR) models to predict pollution levels for urban areas.[48 49] To date, most applications of LUR models for air and noise pollution have been in high-income countries,[13 14 48] with an emerging number in Asian cities and a limited number in African cities.[12 50–52] To generate high-resolution estimates of air and noise pollution in the GAMA, we will build LUR models with spatial and temporal predictor variables. The models will also include terms that allow for the capturing of systematic temporal patterns, for example, random intercepts for hour of the day or month of the year, and terms that use pollution levels at fixed sites to remove weekly temporal changes. The models will use year-long data on $PM_{2.5}$, BC, $NO_2$ and $NO_x$ concentrations, aggregated to weekly average concentrations, and sound level metrics aggregated hourly and daily. The LAeq metric will be modelled hourly so that within-day patterns of sound variation can be captured in the model and then model predictions can be used to construct $LAeq_{24hr}$, $L_{day}$, $L_{night}$ and $L_{den}$. The specific temporal and spatial structures that are built into the models will be determined from the descriptive work.

We will obtain spatial/location-based predictor variables from publicly available sources (eg, OpenStreetMap), government databases, and satellite imagery to collate data on transportation networks (eg, road type), land cover/land use, locations of interest and green and blue

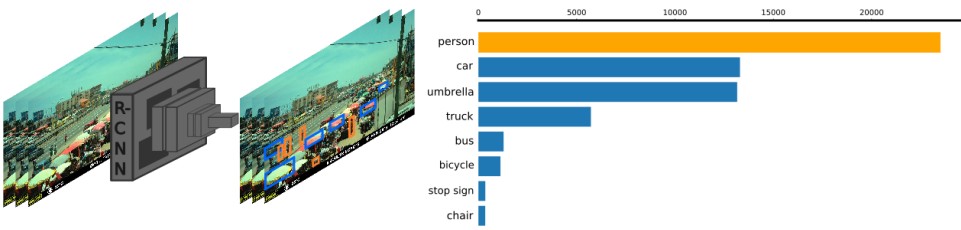

**Figure 6** Illustration of how object detection models and street-level imagery can be combined from the Accra campaign data to identify potential correlates of air and noise pollution in the imagery. Information recorded on the bottom of the images includes the date and time, camera name and the ambient temperature. The numbers illustrate an example of object counts within imagery.

spaces. We also have temporal information on meteorological conditions (eg, temperature, wind speed and direction, humidity) from local weather stations that are colocated with six fixed-site environmental monitors. Appropriate data checks will be done to ensure that model assumptions are met along with 10-fold hold-out cross-validation methods to assess model performance in different parts of the city.

We will be reporting the model results in the form of estimates that represent annual average levels of $PM_{2.5}$, $NO_2$, $NO_x$, $LAeq_{24hr}$, $L_{den}$ and $IR_{24hr}$. We will also provide maps that show estimates that are disaggregated by season (eg, Harrmattan and non-Harrmattan for air pollution) and within day (eg, day vs night).

### Identification of sources of air and noise pollution with imagery and audio

We will glean insights into the determinants and correlates of air and noise pollution (ie, potential sources) in both space and time by applying machine learning approaches, novel in the domain of pollution research, to our time-lapse images collected every 5 min at the ~140 measurement sites.[53] We will use Object Detection algorithms, implemented in a Convolution Neural Network architecture, to identify predefined object classes within an image with a rectangular box bounding their presence (figure 6). We will modify pre-existing algorithms to include custom object classes specific to our study context such as roadside cookstoves and street loudspeakers, and as determined by the research team.[54–57] A sample set of prelabelled images will be used to fine-tune a pretrained object detection algorithm to detect the objects of relevance to this study. The algorithm will then be applied to all images collected during the campaign to produce a list of variables that can be included as independent variables in models estimating the association of air and noise pollution levels with the occurrence of these variables in high spatial–temporal resolution. This approach could be extended to potential future applications such as estimating traffic flows (segmented by vehicle type such as bicycles, cars and minivans whose average emissions vary) or to apply the model to new sources of street level imagery data to identify correlates of air and noise pollution at unmeasured locations across the city.[53]

Similarly, machine learning models can be applied to the audio to classify different sound types and identify sound sources. Some models can predict over 500 different sound types/sources (eg, dog barking, ocean waves, car engine revving) and have been pretrained on 2 million short audio clips.[58 59] The recent wave of development of these models highlights advancements in this field. However, the transferability of the available models, which have predominantly been trained on data from high-income cities and countries,[39 58 60] to a setting such as Accra will have to be tested and understood.

### Air pollution impacts of policy and urban planning

We plan to use deterministic process-based models of air pollution to estimate the air pollution impacts of policies and urban planning decisions in Accra. Process-based models such as meteorological chemical transport and dispersion models[61–63] can provide quantitative estimates of the air pollution impacts of different policy scenarios by modifying sources according to the specific scenario. After minimising errors in meteorological inputs by nudging to ECMWF (European Centre for Medium-Range Weather Forecasts) meteorological reanalysis data, the deterministic relationships between the model's emissions inputs and concentration outputs will be used in conjunction with the measurement data to calibrate the highly uncertain SSA emissions data. This relationship will be recreated using Gaussian process emulation[64] to simulate the millions of model runs required for a Bayesian Monte Carlo calibration[65] exercise, in which each run is weighted according to its output's agreement with the measurements. The same weights are applied to the corresponding emissions inputs, producing a distribution of emissions values, the modal value of which is taken as the calibrated input. Repeating this at multiple model time steps averages the calibration over the values of the many other varying model inputs. The remaining measurements will then be used to validate the model's outputs, after it is rerun with the calibrated emissions. Following validation, the model (if appearing to perform well) can be used for ongoing policy and urban planning scenario testing exercises for emissions reduction policies in Accra, and other SSA cities with similar source profiles.

### PATIENT AND PUBLIC INVOLVEMENT

No patients or members of the public were involved in this component of the study.

## ETHICS AND DISSEMINATION

This environmental study was deemed exempt from full ethics review at Imperial College London and the University of Massachusetts Amherst; it was approved by the University of Ghana Ethics Committee. While pollution sources (cars, roadside cookstoves and loudspeakers, etc) are the targets of our field camera and audio recorders, bystanders in public places and their voices may sometimes be in the mix. Monitors are placed at about 4 m height where faces are normally not recognisable in the images and conversations unintelligible in the audio. Further, the audio recorders record for only 10 s every 10 min. Extra precautions (eg, blurring of faces in imagery) is taken to maintain privacy of bystanders.

Both public and private stakeholders and relevant civil society groups will be invited to annual research consortium meetings where preliminary and final results will be shared. This will enable policy-makers to frame and understand impacts of current and future policy scenarios. Additionally, results will be presented at international conferences and published in peer-reviewed journals. Further, we will also engage with civil society through blog posts and other social media platforms.

**Author affiliations**
[1]Department of Epidemiology and Biostatistics, Imperial College London, London, UK
[2]MRC Center for Environment and Health, Imperial College London, London, UK
[3]Department of Environmental Health Sciences, University of Massachusetts Amherst, Amherst, Massachusetts, USA
[4]School of Population and Public Health, The University of British Columbia, Vancouver, British Columbia, Canada
[5]Abdul Latif Jameel Institute for Disease and Emergency Analytics, Imperial College London, London, UK
[6]Regional Institute for Population Studies, University of Ghana, Legon, Accra, Ghana
[7]Institute for Health and Social Policy, McGill University, Montreal, Quebec, Canada
[8]Department of Epidemiology, Biostatistics, and Occupational Health, McGill University, Montreal, Quebec, Canada
[9]Department of Physics, University of Ghana, Legon, Accra, Ghana
[10]Department of Environmental Health, Harvard T.H. Chan School of Public Health, Boston, Massachusetts, USA
[11]Department of Geography and Resource Development, University of Ghana, Legon, Accra, Ghana
[12]NIHR HPRU in Environmental Exposures and Health, Imperial College London, London, UK

**Contributors** All the authors contributed to this work and have taken part in the academic discussion for writing the study protocol, drafting the article and revising it. SNC, ASA, MB, ME, JB, MBT, AFH, JM, ST, JN, JV, SA-M, EA, BB and RA gave substantial contributions to conception and design and acquisition of data. SNC, ASA, MB, ME, MBT, RN, EM, JG, JW, AB, FK, SB and RA gave substantial contributions to the analysis plan for data. SNC, ASA, MB, ME, JB and RA drafted and revised the manuscript. All authors reviewed the final version.

**Funding** This work is supported by the Pathways to Equitable Healthy Cities grant from the Wellcome Trust (209376/Z/17/Z). SC is supported by a Canadian Institutes for Health Research PhD scholarship as well as an Imperial College President's PhD scholarship.

**Disclaimer** The content of this manuscript is solely the responsibility of the authors and does not necessarily represent the official views of the funders.

**Map disclaimer** The depiction of boundaries on this map does not imply the expression of any opinion whatsoever on the part of BMJ (or any member of its group) concerning the legal status of any country, territory, jurisdiction or area or of its authorities. This map is provided without any warranty of any kind, either express or implied.

**Competing interests** None declared.

**Patient consent for publication** Obtained.

**Provenance and peer review** Not commissioned; externally peer reviewed.

**ORCID iD**
Sierra N Clark http://orcid.org/0000-0002-8592-3466

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
