## [Reviewer comments · BMJ Open]

ARTICLE DETAILS

TITLE (PROVISIONAL)	High-resolution spatiotemporal measurement of air and environmental noise pollution in sub-Saharan African cities: Pathways to Equitable Health Cities Study protocol for Accra, Ghana
AUTHORS	Clark, Sierra; Alli, Abosedo; Brauer, Michael; Ezzati, Majid; Baumgartner, Jill; Toledano, Mireille; Hughes, Allison; Nimo, James; Moses, Josephine; Terkperter, Solomon; Vallarino, Jose; Agyei-Mensah, Samuel; Agyemang, Ernest; Nathvani, Ricky; Muller, Emily; Bennett, James; Wang, Jiayuan; Beddows, Andrew; Kelly, Frank; Barratt, Benjamin; Beevers, Sean; Arku, Raphael

VERSION 1 – REVIEW

REVIEWER	Daniel Shepherd Auckland University of Technology Auckland New Zealand
REVIEW RETURNED	19-Jan-2020

GENERAL COMMENTS	Review of : High-resolution assessment of air and environmental noise pollution in sub-Saharan African cities: Pathways to Equitable Health Cities Study protocol for Accra, Ghana This paper describes the research protocols, but not the outcomes of, a study planned in the Ghanaian city of Accra. The authors have formed a multinational team that provides expertise in all aspects of the study. As presented the study is exciting and its design well considered. The points below are suggestions only, and which the authors can take-or-leave: 1) Check that acronyms are defined prior to use, and don't overuse them. For example, the acronym CNN is used only once, so just use the full term in both its occurrences.2) First sentence of the introduction: "...with the number of urban dwellers having increased by over 400% since 1980...". Can the absolute numbers be provided to set the scene and give an impression on how populated this region actually is?3) Throughout. This is a submission to the British Medical Journal, though American English appears to have been used (e.g., liter vs litre). Can the spell checker be changed?4) Abbreviations, perhaps hour and hours instead of hr and hrs.
--

	5) LAeq is an aggregate measure that can sometimes average out important sound events of interest. When analysing the data will the authors also be looking at a maximum or peak value? If not then this needs to be justified. 6) No human survey is being undertaken, and this I assume is due to budgetary constraints. Shame as this is a missed opportunity. Do the authors intend to access medical/hospital records during the recording epochs or retrospectively, and if so has this access been arranged? 7) By-and-large the manuscript is well written, though the odd gremlin was noted (e.g., "...and rugged time-lapse camera..." should be "...cameras...") and one final proofread would be useful. I wish the authors all the best with their ambitious and well thought-out study.
--	---

REVIEWER	Jeffrey R. Brook Dalla Lana School of Public Health University of Toronto CANADA
REVIEW RETURNED	01-Mar-2020

GENERAL COMMENTS	A. Measurement Network Design Four classes of land cover resolved at 20x20m have been defined. - Can the authors clarify that a possible location is anywhere within the selected 20x20m area depending upon practicalities. - Can the authors indicate what criteria are set when a "suitable spot to the ideal location is identified as a replacement" because the randomly identified 20x20m area is not feasible? As each set of five rotating sites is selected through the stratified random design based upon land use type what is considered for geographic coverage. There are two logical considerations: 1) Select the five in close enough proximity to make the set up efficient given time limitations and likely time loss in traversing the region and potentially benefit from having simultaneous sampling within smaller portions of the city (i.e., study of neighbourhood scale variability) 2) Require the five to be geographically spread given that there are potential microclimate and source-related differences present across the region that a more-dispersed set of five sites will better capture. Readers of the protocol could benefit from the author's perspective and balancing of these two possible options (or others) and ultimately whether these were considered in the design. B. Section on: QA/QC QA/QC are critical to establish and the team have stated their approach of duplicates, blanks, and period co-location at reference locations. It would be helpful for them to further state (a priori) what are the required levels of accuracy and precision and contamination and what steps are taken when they are not achieved. Data rejection? Data adjustment with external information? C. Section on: High-resolution estimates of air and noise pollution in the GAMA
--

	The authors indicate that spatially and temporally resolved estimates are of interest. Variability occurs on many temporal scales and due to multiple physical processes. Overall, more information on the primary spatial-temporal modelling approach planned would be useful to readers of the protocol. Spatial maps of chronic exposure are also a common output of this research. Based upon the temporal resolutions of the different measurements and the necessary rotating design of the campaign where at most there are 15 sites operating simultaneously, with unspecified geographic spread for 1/3 of the sites (see comment in A above). Given this, the protocol would benefit from more details of the expected data analysis approach to isolate the temporal scales of interest and to also temporally adjust the measures across the 130 locations, given the rotation in locations every week, to separate spatial differences due to location versus week of measurement. D. Page 19: “Applying such methods to the images can create a temporal catalogue of a diverse set of objects present each location, which could then be used to predict and model pollutant levels in both space and time” - Comparable imagery are needed at unmeasured locations in order to predict using these models. What is the intention here? How will these images be acquired and at what sort of coverage (e.g., spatial resolution) will this occur? E. Page 19: “to calibrate the highly uncertain SSA emissions input data.” - Is there a methodology of calibration that will be applied? Are there particular and possibly SSA-unique emission sources that will be the focus given the large number of sources and likely varying uncertainties and the many degrees of freedom in doing emission sensitivity analyses? - In addition to an improved and validated CTM, notwithstanding that lack of agreement between observations and predictions are also due to uncertainties in the simulation of meteorology, which must be considered, will the presumed-improved emission inventory be independently validated? Will this inventory be made publicly available?
--	---

VERSION 1 – AUTHOR RESPONSE

Reviewer 1

1) Check that acronyms are defined prior to use, and don’t overuse them. For example, the acronym CNN is used only once, so just use the full term in both its occurrences.

We have carefully checked our use of acronyms throughout the manuscript, and when used, terms are defined at first use. We have also reduced the use of acronyms to those that are relatively common in the air pollution and noise files.

2) First sentence of the introduction: “...with the number of urban dwellers having increased by over 400% since 1980...”. Can the absolute numbers be provided to set the scene and give an impression on how populated this region actually is?

We have added additional information to the first sentence on the number of urban dwellers in SSA and stated that:

“Sub-Saharan Africa (SSA) is the world’s fastest urbanising region, with the number of urban dwellers having increased by over 400% from 84 million in 1980 to an estimated urban population of ~450 million people in 2020 [1].” (p. 5).

3) Throughout. This is a submission to the British Medical Journal, though American English appears to have been used (e.g., liter vs litre). Can the spell checker be changed?

Many thanks for highlighting this. We have changed to British English.

4) Abbreviations, perhaps hour and hours instead of hr and hrs.

Hr/hrs has been changed to hour/ hours.

5) LAeq is an aggregate measure that can sometimes average out important sound events of interest. When analysing the data will the authors also be looking at a maximum or peak value? If not then this needs to be justified.

We agree that metrics that capture episodic sound events should be considered. We had indicated that, in addition to LAeq, we will compute the Intermittency Ratio (IR), a metric which captures event-based sound and represents the percentage of sound from discrete sound-events over a specified time interval (typically 24hr, day-time or night-time). We will also use maximum noise level (LA_{max}) to characterize the maximum sound level of discrete events. We have revised the manuscript to highlight our use of different noise metrics:

“We will provide summary statistics and visuals of the spatial and temporal patterns (within- and between-day, and seasonal) of air pollution ($PM_{2.5}$, NO_2) concentrations and average-based metrics of noise pollution such as $LAeq_{24hr}$, daytime (L_{day}), nighttime (L_{night}), and day-evening-night weighted L_{den} . Additionally, we will include metrics which capture short-term and episodic sound events such as the average maximum sound level and a novel metric that captures the percentage of event-based sound (the Intermittency Ratio (IR_{24hr} , IR_{day} , IR_{night})) [44].” (p. 16/17)

6) No human survey is being undertaking, and this I assume is due to budgetary constraints. Shame as this is a missed opportunity. Do the authors intend to access medical/hospital records during the recording epochs or retrospectively, and if so has this access been arranged?

Due to resource constraints and in an effort to manage this process, we focused our attention at the present time on getting the environmental pollution data collection right. However, the larger *Pathways to Equitable Healthy Cities* project has other research aims, that include both characterising exposure of specific demographic subgroups and collating demographic and health records. As the Reviewer and Editors are inevitably aware, even national vital registration system such as deaths and births are incomplete in much of sub-Saharan Africa and typically not held in central repositories. Therefore, the identification, collation, and evaluation of the completeness of health data would be a distinct process, with its own methodological components, and far beyond the scope of an environmental measurement campaign which the current paper describes.

7) By-and-large the manuscript is well written, though the odd gremlin was noted (e.g., “...and rugged time-lapse camera...” should be “...cameras...”) and one final proofread would be useful.

We have conducted final proof-reads and hope that we have caught any straggling grammatical errors. We of course welcome Editorial input into the final manuscript to enhance readability.

I wish the authors all the best with their ambitious and well thought-out study

Many thanks!

Reviewer: 2

A. Measurement Network Design

Four classes of land cover resolved at 20x20m have been defined.
- Can the authors clarify that a possible location is anywhere within the selected 20x20m area depending upon practicalities.

During the computer sampling of locations, pre-defined lat/long coordinates (points) were returned as the target measurement site locations, as opposed to a 20m raster grid cell. The field team used the coordinates of these target locations to find suitable measurement sites as near as possible to the computer-generated locations and with the same land use characteristics. This flexibility and practicality allow us to find suitable locations that meet the needs of our sampling design and data, within practical constraints of field work. As the campaign goes on, we continually review the balance of the numbers of actual measurement sites that fall within the land use strata we used for the original sampling and will actively sample additional sites to keep the design balanced if needed. We have added additional information on our site selection procedure to the manuscript and those changes.

“In selecting the rotating site locations, we used a stratified random sampling approach:

1. *The study area (GAMA) was stratified by a land use grid (20m x 20m raster converted into a polygon shapefile) with four classes (medium/ low-density residential, high-density residential, commercial, business, and industrial areas, and ‘other’ areas (e.g. parks, forest, agricultural areas)) [31] and inside or outside the main Accra Metropolitan Area (AMA).*
2. *The computer then generated and returned the latitude/ longitude coordinates of a random sample of 130 target measurement site locations within strata.*
3. *Target measurement locations were first examined by overlaying point locations onto Google Maps and Google Earth to identify sites that were in restricted areas (e.g., military barracks). Sites in restricted areas were re-sampled to a nearest suitable spot that also fell within the same type of land use strata (n=~5 sites).*
4. *Using the coordinates of the target sampling locations, the field team then visit individual sites throughout the campaign to find measurement sites at or as close as possible to the target locations and also with the same land use characteristics.*
5. *When a site is deemed structurally sound for the field team to install equipment at (e.g., staircase to the roof) and can allow for the equipment to be installed at a target height, permission is requested from the site owner/ manager (more details on the logistics of field work are in sections below).*
6. *During the course of the measurement campaign, we will actively review the balance between the number of actual measurement sites by land use strata as originally designed, and potentially sample additional sites to make up for unrepresentative site types.”*

(p. 8)

- Can the authors indicate what criteria are set when a “suitable spot to the ideal location is identified as a replacement” because the randomly identified 20x20m area is not feasible?

As above, when the computer-generated site was deemed not feasible due to its placement in restricted or implausible areas (e.g., military barracks, in the airport), a nearest suitable spot that also fell within the same type of land use strata to the ideal location was identified as a replacement. This was done prior to the field team going out to locations and searching for actual measurement sites. This has been expanded on in the manuscript, page 7:

“Target measurement locations were first examined by overlaying point locations onto Google Maps and Google Earth to identify sites that were in restricted areas (e.g., military barracks). Sites in restricted areas were re-sampled to a nearest suitable spot that also fell within the same type of land use strata (n=~5 sites)” (p. 8).

As each set of five rotating sites is selected through the stratified random design based upon

land use type what is considered for geographic coverage. There are two logical considerations:

1) Select the five in close enough proximity to make the set up efficient given time limitations and likely time loss in traversing the region and potentially benefit from having simultaneous sampling within smaller portions of the city (i.e., study of neighbourhood scale variability)

2) Require the five to be geographically spread given that there are potential microclimate and source-related differences present across the region that a more-dispersed set of five sites will better capture.

Readers of the protocol could benefit from the author's perspective and balancing of these two possible options (or others) and ultimately whether these were considered in the design.

Scheduling sampling at the rotating sites generally followed option 1, with the primary goal of enhancing the efficiency of our fieldwork set-up. Since the GAMA area is quite large (1500 km²) and traffic in and out of the city can result in grid-lock for hours, the team chooses four to five sites that are in proximity to each other or are within the same geographic area from the list of possible locations to monitor. Nonetheless, for some of the rural background ('other') areas, the sites could actually be geographically quite far away from each other. While selecting four to five locations that were geographically spread out or at least were diverse in their land use classifications would have allowed capturing a range of microclimate and source related differences within each week, it was not logistically feasible. To partially overcome this, we instructed the team to sample at new geographic areas/land use types from one week of monitoring to another (e.g., week 1 set of 4-5 sites: north west of GAMA background areas; week 2 set of 4-5 sites: south east of GAMA high-density residential and low-density areas). Further, during analysis, we will use data from the fixed sites to account for (remove) time trend in the data from the rotating sites.

We have expanded on these considerations in the manuscript in the first paragraph on page 13:

"For the rotating sites, 4-5 locations are monitored each week. Because of logistical and time constraints related to setting up each site, the team chooses sites that are within the same part of the city, but may have varying land use characteristics (e.g., mix of low and high-density residential locations). Monitors are retrieved seven days after initiating the measurements for data download and equipment cleaning in the field laboratory. The monitors are then re-deployed 48 hours later at a new set of locations in a different geographic area, with the aim of capturing potential microclimate and source-related differences between areas which likely impact pollution." (p. 13).

B. Section on: QA/QC
QA/QC are critical to establish and the team have stated their approach of duplicates, blanks, and period co-location at reference locations. It would be helpful for them to further state (a priori) what are the required levels of accuracy and precision and contamination and what steps are taken when they are not achieved. Data rejection? Data adjustment with external information?

We have added further details to the manuscript on QA/QC regarding our *a priori* decisions of precision/accuracy and contamination cut-offs, and data rejection and adjustment. Due to journal word count restraints, we have moved the entire section on QA/ QC to the SI materials.

"Throughout the campaign, we will follow a set of procedures and protocols to uphold and assess the quality of the data being generated. We follow the principles that all procedures should be carefully planned, tested, and performed, the origin and life-course of all data must be traceable, and any deviations or irregularities must be recorded. Throughout all data collection, documentation of sampling and conditions will be maintained in field notebooks. Furthermore, data collection logs will be digitized and backed up electronically on hard-drives and an online server, which will be checked on a daily basis for accuracy. The field team were given multiple weeks of project specific training prior to the pilot measurements commencing. The team were taught specific protocols for equipment handling and cleaning, data inspection and cleaning, and equipment installation at measurement sites. The team were also given hardcopies of the protocols and, in addition to field visits by researchers, had constant remote access via phone/ web to project researchers throughout the campaign. In the SI, we have included further information on our precision and accuracy testing, protocol for blank and duplicate collection, and data cleaning and inspecting procedures (SI 1)". (p.15/16)

(SI 1 TEXT): “The field team calibrate equipment prior to each use. Specifically, the UPAS mass flow sensor maintains a steady sampling flow rate over time by internally measuring changes in pressure drop across the filter media. But as part of our quality assurance process, the flow rates are manually checked with a TSI Mass Flowmeter (4000 Series) for possible flow drift prior to and immediately after each monitoring session. Monitors are adjusted as necessary prior to the next deployment. Following a previous protocol used in the same setting [1], samples will be considered valid only if the average flow rate is within 10% of the intended rate of 1 lpm, and the UPAS operated for $\geq 85\%$ of the 7-day measurement period. Additionally, the SLMs are calibrated prior to each monitoring session with a CA114 sound calibrator at $94.0 \text{ dB} \pm 0.3 \text{ dB}$ and $1000\text{Hz} \pm 0.5\%$ (Convergence Instruments, Canada). If an instrument is consistently reading a calibration offset $\pm 1 \text{ dBA}$, the SLM is pulled out of commission and tested and the data from that session considered invalid.

In order to understand the extent of potential filter and diffusion pad contamination from handling procedures, we collect field blanks at 20% of our sites for filter based $\text{PM}_{2.5}$ and NO_x and NO_2 samples. Blank $\text{PM}_{2.5}$ samples are prepared as regular samples in the field lab, brought to the field sites, and deployed in the same way as the regular sample, but without the pump being turned on. NO_x/NO_2 blanks are brought to the field sites but not exposed to air in their sealed canisters. During analysis, information from the blank samples will be used to account for residual contamination from the laboratory work, transportation, and field handling processes, which in a past study in Accra was minimal [1]. We will assess the mean absolute difference of the pre- and post-sampling weights of the blank samples; mean weights within 10 ug will be considered valid [1]. Also, final filters weights will be checked against the limit of detection, computed using the blanks, to be sure all valid samples are above this limit.

We will assess the accuracy and precision of our monitors by conducting **pre-campaign** side-by-side monitoring sessions between all our instruments of the same type (precision) and our instruments next to reference grade or higher-grade monitors (accuracy).

- Prior to field deployment, we tested minute-by-minute monitor-monitor precision for the continuous $\text{PM}_{2.5}$ monitors by running all of our monitors alongside each other over a 24-hour period at the University of Ghana, Legon campus, with average relative humidity (RH) ($\sim 78\%$) and temperature ($29 \text{ }^\circ\text{C}$) representative of the city. The continuous $\text{PM}_{2.5}$ measurements had good agreement and were within $2\text{-}3 \text{ ug}/\text{m}^3$ of each other. The continuous $\text{PM}_{2.5}$ ZeFan monitor uses the Plantower sensor (model PMS7003) which has been validated in previous studies against a TEOM 1400a analyser and tested for durations ranging from 6 months to a year in various environmental conditions [2,3].
- The filter-based UPAS monitor has been evaluated in previous laboratory and field settings against a federal reference monitor (URG-2000-30EGN-A; URG Corp., USA), personal environmental monitor (PEM 761-203; SKC, Inc., USA) and Harvard Impactors, respectively and has proven valid for ambient, household, and personal monitoring in a typical tropical climate as our study [4–6].
- Our pre-campaign tests of SLM monitor-monitor precision showed good agreement. There was only a 0.5 dBA difference between the monitoring period median values ($\text{LAeq}1\text{min}$) for 50% of monitors within the IQR bounds around the overall median (25%-75%) and a 1.7 dBA difference between the two monitors with the highest and lowest monitoring period median values. The monitor-monitor precision test was done in Accra and SLMs were exposed 16hrs to multiple sound environments similar to what we would expect during the full monitoring campaign. Our Type II Noise Sentry SLMs were also validated in a separate aircraft noise study conducted in San Francisco against a Type I industry standard instrument (DUO 01dB) [7], and the agreement was high (mean and median second by second difference between the instruments was -0.42 and -0.38 dBA , respectively).

In addition to the pre-campaign monitor-monitor precision tests and accuracy checks, we will collect duplicate samples at 20% of our sites and conduct **mid and post-campaign** precision tests to check their sensitivity over time and accuracy checks with reference grade monitors.

- To understand the extent to each type of monitor provides consistent measurements among all the units used in the campaign, we are also collecting duplicate samples from co-located instruments at 20% of our rotating measurement sites. Duplicate samples will be evaluated from 20% of sites during the course of the campaign and faulty and malfunctioning instruments

will be pulled from the field and data potentially removed from analysis if mean absolute difference between duplicate measurement is $> 10 \text{ ug/m}^3$ [1] or $>2 \text{ dBA}$ ($\text{LAeq}_{24\text{hr}}$).

- We will additionally co-locate all of our monitors side-by-side for mid and post campaign precision tests for a 1-week period to assess instrument drift over time. Data will be considered invalid if the mean absolute difference between daily/ weekly $\text{PM}_{2.5}$ and $\text{LAeq}_{24\text{hr}}$ measurements differ by $> 10 \text{ ug/m}^3$ [1] or $>2 \text{ dBA}$.
- Since light-scattering techniques only infer PM mass from detecting particle number concentrations and are impacted by weather conditions (i.e. RH and temperature), their estimates of mass concentration are inexact. Thus, we will co-locate the ZeFan monitors with a U.S. federal equivalent continuous monitor Met One BAM 1020 at three sites, each with unique source influence in Accra for a week at the end of the campaign and adjust the minute-by-minute continuous PM records for impact of RH and then their average against the co-located integrated $\text{PM}_{2.5}$ concentrations from UPAS.

The real-time data will be inspected weekly by the field team as it is downloaded from the instruments. Potential implausible values will be identified by inspecting all values that are 5-standard deviations above or below the site and day (or week for filter-based $\text{PM}_{2.5}$ and NO_x/NO_2) specific mean value. For the filter based $\text{PM}_{2.5}$ data, potentially implausible values will be checked against the monitor run time, weighed mass value, and flow rate. The log sheets will be checked to see if any information on instrument malfunction or other irregularities was noted for the continuous $\text{PM}_{2.5}$ and SLM monitors. Values deemed erroneous will be dropped from analysis. Additionally, since monitors are swapped every week, sometimes an entire week of data might be erroneous if the instrument is malfunctioning or if calibration did not occur correctly. We will identify outlier weeks by plotting timeseries of a month worth of data to identify any potential implausible weeks of data and conduct instrument checks, review log sheets, and drop or correct data as needed. Finally, all real-time instruments will have their first 5 minutes of data dropped to allow the instruments to stabilize and the data further trimmed to match the exact monitoring session start and end date and time as recorded by the field team on the data log forms.”

C. Section on: High-resolution estimates of air and noise pollution in the GAMA

The authors indicate that spatially and temporally resolved estimates are of interest. Variability occurs on many temporal scales and due to multiple physical processes. Overall, more information on the primary spatial-temporal modelling approach planned would be useful to readers of the protocol.

Prior to model building, we will use descriptive statistics and data visualizations to examine variations and patterns of air and noise pollution metrics over space (e.g., distance), type of place (e.g., land use) and multiple time scales (within day, between day, and seasonally). We have added more information on this pre-modelling work in the paper.

“We will provide summary statistics and visuals of the spatial and temporal patterns (within- and between-day, and seasonal) of air pollution ($\text{PM}_{2.5}$, NO_2) concentrations and average-based metrics of noise pollution such as $\text{LAeq}_{24\text{hr}}$, daytime (L_{day}), nighttime (L_{night}), and day-evening-night weighted L_{den} . Additionally, we will include metrics which capture short-term and episodic sound events such as the average maximum sound level and a novel metric that captures the percentage of event-based sound (the Intermittency Ratio ($\text{IR}_{24\text{hr}}$, IR_{day} , IR_{night})) [44].” (p .16/17)

The spatial and temporal modelling approach will be informed by these prior explorations and will likely differ between air and noise pollution.

We have expanded on how we will develop our modelling approaches.

“To generate high resolution estimates of air and noise pollution in the GAMA, we will build LUR models with spatial and temporal predictor variables. The models will also include terms that allow for the capturing of systematic temporal patterns, e.g. random intercepts for hour of the day or month of the year, and terms that use pollution levels at fixed sites to remove weekly temporal changes. The models will use year-long data on $\text{PM}_{2.5}$, BC, NO_2 , and NO_x concentrations, aggregated to weekly average concentrations, and sound level metrics aggregated hourly ($\text{LAeq}_{1\text{hr}}$) and daily ($\text{IR}_{24\text{hr}}$). The LAeq metric

will be modelled hourly so that within-day patterns of sound variation can be captured in the model and then model predictions can be used to construct LA_{eq24hr} , L_{day} , L_{night} , and L_{den} . The specific temporal and spatial structures that are built into the models will be determined from the descriptive work.” (p.17)

..... “Appropriate data checks will be done to ensure that model assumptions are met along with 10-fold hold-out cross validation methods to assess model performance in different parts of the city. Possible spatial autocorrelation in the data will be investigated by generating variogram plots of the raw data and the model residuals.” (p.17/18)

We are also including, or conducting sensitivity tests with, temporal and spatial predictor variables in our models. For instance, we are including hourly, daily, or weekly aggregated weather variables to the pollution models as well as scraping a variety of sources (e.g., OpenStreetMap) for robust datasets on the presence and locations of key spatial predictor variables.

We have expanded with some additional points.

“We will obtain spatial/ location-based predictor variables from publicly available sources (e.g. OpenStreetMap), government databases, and satellite imagery to collate data on transportation networks (e.g., road-type), land cover/land use, points of interest (e.g., traffic lights, restaurants), and green and blue spaces. We also have temporal information on meteorological conditions (e.g., temperature, wind speed and direction, humidity) from local weather stations that co-located with 6 fixed-site environmental monitors.” (p.17)

Spatial maps of chronic exposure are also a common output of this research.

We thank the reviewer for this comment and have added mention of this to the manuscript.

“We will be reporting the model results in the form of estimates that represent annual average levels of $PM_{2.5}$, NO_2 , NO_x , LA_{eq24hr} , L_{den} and IR_{24hr} . We will also provide maps that show estimates that are disaggregated by season (e.g. Harmattan and non-Harmattan for air pollution) and within day (e.g., day vs night).” (p. 18)

Based upon the temporal resolutions of the different measurements and the necessary rotating design of the campaign where at most there are 15 sites operating simultaneously, with unspecified geographic spread for 1/3 of the sites (see comment in A above). Given this, the protocol would benefit from more details of the expected data analysis approach to isolate the temporal scales of interest and to also temporally adjust the measures across the 130 locations, given the rotation in locations every week, to separate spatial differences due to location versus week of measurement.

Please refer to our previous responses to comment on item C above. In summary, we will be adjusting for the varying time scales where different measurements were conducted (if it appears needed from the initial descriptive work) within the models with fixed or random terms representing these various time scales.

**D. Page 19:
“Applying such methods to the images can create a temporal catalogue of a diverse set of objects present at each location, which could then be used to predict and model pollutant levels in both space and time”
- Comparable imagery are needed at unmeasured locations in order to predict using these models. What is the intention here? How will these images be acquired and at what sort of coverage (e.g., spatial resolution) will this occur?**

We have clarified in the main text that our present goal is to model and estimate the associations of potential sources of air and noise pollution identified within our collected imagery (e.g., animals, cars, outdoor cooking) with air and noise pollution levels over fine temporal scales (we have images every 5 minutes) and over space (using information on land use and geographic locations). Extending a model

to such an application (e.g. using available street level imagery) could be done and we might explore that avenue in the future.

The main manuscript has been revised to clarify the intention for the image-related modelling (p. 18):

“.....The algorithm will then be applied to all images collected during the campaign to produce a list of variables that can be included as independent variables in models estimating the association of air and noise pollution levels with the occurrence of these variables in high spatial-temporal resolution. This approach could be extended to potential future applications such as estimating traffic flows (segmented by vehicle type such as bicycles, cars and minivans whose average emissions vary) or to apply the model to new sources of street level imagery data, to identify correlates of air and noise pollution at unmeasured locations across the city [53].” (p.18)

E. Page 19:
“to calibrate the highly uncertain SSA emissions input data.”
- Is there a methodology of calibration that will be applied? Are there particular and possibly SSA-unique emission sources that will be the focus given the large number of sources and likely varying uncertainties and the many degrees of freedom in doing emission sensitivity analyses?

- In addition to an improved and validated CTM, notwithstanding that lack of agreement between observations and predictions are also due to uncertainties in the simulation of meteorology, which must be considered, will the presumed-improved emission inventory be independently validated? Will this inventory be made publicly available?

Some extra detail with references to published work describing the key parts of the method has been added to the text. Given that this is a measurement protocol paper, and this section is intended to describe some of the wide-ranging uses that the outputs of the measurement campaign will be put to, space constraints do not allow a full description of the technical details. Whilst we acknowledge that there will be limitations to this method in terms of its ability to isolate individual emissions sources, this must be viewed in light of the fact that current emissions inventories in this part of the world vary by a factor of two or more, so even narrowing the range of plausible emissions totals would be a worthwhile contribution. We are also generating, through our collection and analysis of street level imagery, good temporal (and spatial) information on traffic levels (including differentiating between cars/trucks/lorries/motorcycles), outdoor cooking prevalence and cooking times, among other useful emissions source information. As well, through our modelling and comparison to measurements, we may be able to refine emissions factors.

We have added details on the CTM evaluation and emissions calibration process to the manuscript as indicated below.

“We plan to use deterministic process-based models of air pollution to estimate the air pollution impacts of policies and urban planning decisions in Accra. Process based models such as meteorological chemical transport and dispersion models [61–63] can provide quantitative estimates of the air pollution impacts of different policy scenarios by modifying sources according to the specific scenario. After minimizing errors in meteorological inputs by nudging to ECMWF meteorological re-analysis data, the deterministic relationships between the model’s emissions inputs and concentration outputs will be used in conjunction with the measurement data to calibrate the highly uncertain SSA emissions data. This relationship will be recreated using Gaussian process emulation [64] to simulate the millions of model runs required for a Bayesian Monte Carlo calibration [65] exercise, in which each run is weighted according to its output’s agreement with the measurements. The same weights are applied to the corresponding emissions inputs, producing a distribution of emissions values, the modal value of which is taken as the calibrated input. Repeating this at multiple model time-steps averages the calibration over the values of the many other varying model inputs. The remaining measurements will then be used to validate the model’s outputs, after it is re-run with the calibrated emissions. Following validation, the model (if appearing to perform well) can be used for ongoing policy and urban planning scenario testing exercises for emissions reduction policies in Accra, and other SSA cities with similar source profiles. (p.19)

Finally, we plan to make this inventory publicly available.

VERSION 2 – REVIEW

REVIEWER	Daniel Shepherd Auckland University of Technology Auckland New Zealand
REVIEW RETURNED	19-Apr-2020
GENERAL COMMENTS	Thank you for addressing my concerns. All the best with your study.